# Understanding the Office: Using Ecological Momentary Assessment to Measure Activities, Posture, Social Interactions, Mood, and Work Performance at the Workplace

**Lina Engelen** [1,2,*] and **Fabian Held** [3]

1   Charles Perkins Centre and Prevention Research Collaboration, Sydney School of Public Health, University  of Sydney, Sydney 2006, Australia
2   School of Health and Society, University of Wollongong, Wollongong 2522, Australia
3   Charles Perkins Centre, University of Sydney, Sydney 2006, Australia; fabian.held@sydney.edu.au
*   Correspondence: lina.engelen@sydney.edu.au

**Abstract:** Studying the workplace often involves using observational, self-report recall, or focus group tools, which all have their established advantages and disadvantages. There is, however, a need for a readily available, low-invasive method that can provide longitudinal, repeated, and concurrent in-the-moment information to understand the workplace well. In this study, ecological momentary assessment (EMA) was used to collect 508 real-time responses about activities, posture, work performance, social interactions, and mood in 64 adult office workers in three Australian workplaces. The response rate was 53%, and the time to fill out the survey was 50 seconds on average. On average, the participants were sitting, standing, and walking in 84%, 9%, and 7% of survey instances, respectively. The participants reported they were working alone at their desks in 55% of all reported instances. Reported mood varied up to nine points within one person over the course of the post-occupancy observations. EMA can be used to paint a rich picture of occupants' experiences and perceptions and to gain invaluable understanding of temporal patterns of the workplace, how the space is used, and how aspects of the workplace interact. This information can be used to make improvements to the physical and social workspaces and enhance occupants' work performance and mood.

**Keywords:** EMA; perceptions; occupants; evaluation; office

## 1. Introduction

Close to 80% of Australia's workforce is employed in service and related industries [1], and many of these workers are based in offices. To support the activities performed in an office environment, organizations try to create purposeful workplaces with the aim of increasing work performance, wellbeing, staff perceptions, and retention (e.g., Google's office designs that promote "casual collision" of people and ideas or Microsoft's Milan office, which features personalized communal spaces [2]). The organization of the building, physical arrangement of employees' immediate work area, ambient environmental qualities, exterior amenities, and site planning have all been examined with regard to their health relevance [3]. In order to effectively deliver this support, it is important to understand the workplace, the activities taking place, the social interactions, and perceptions. Office work is often considered synonymous with using a computer while sitting at a desk [4]; however, most people's workday consists of a combination of various activities or tasks, such as concentrated work, scheduled and ad hoc meetings, breaks, and transit, in various proportions. However, getting a good measure of

the proportion of time workers spend in these activities, how they feel, and how they engage with others is not a straightforward task. Many approaches involve observational methods that can be costly and are often only a snapshot of what is happening.

Consequently, scant in-depth information is available on how office spaces are used and how they affect users' behavior, work performance, and perceptions. Information about office workers' perception of their creativity, work performance, and mood has previously been based on one-off post-occupancy survey ratings [5,6] or qualitative focus groups and interviews. The former can be coupled with biases, such as recall bias, and it typically only provides information about one point in time or a composite perception [7]. Although the latter is rich in information, it is a time-consuming, invasive, and expensive way to collect data and therefore infeasible to be used on a large scale. Hence, it is desirable to explore alternative methods to collect more detailed data but with relatively low burden on the participant.

Data collection with ecological momentary assessments (EMAs) is a contender for effectively elucidating these effects and interactions. EMAs have their roots in psychology [8,9] and sample a range of perceptions and experiences simultaneously in real time. Originally, the method was delivered through pen and paper, then progressed through personal digital assistants (PDAs), and then to smartphones today (e-EMA) [5]. e-EMA works through mobile apps that prompt participants to respond to a short survey multiple times per day. The merit of e-EMA is the ability to collect in-the-moment responses on several aspects concurrently in a quick and relatively nonintrusive manner over long or short periods of time. Using this method, it is possible to get simultaneous responses of posture, activities, work performance, and wellbeing from the same person at various times of the workday. The e-EMA responses are time-stamped, which allows for temporal analyses of perceptions and responses. Despite their versatility and ease of use, they have only seldom been used in workplace settings [10]. In a previous pilot study, the method was deemed acceptable and sensitive for use in workplace health evaluations [11], but a larger sample in a range of workplaces is necessary to determine its usefulness in an office setting.

The aims of this study were to ascertain if the e-EMA method is useful to characterize the office workplace in terms of measuring work activity, posture, social interactions, mood, and work performance in the office environment and to understand how these are related.

## 2. Materials and Methods

This study is a secondary analysis of data collected in three studies as part of the "Move to New Building" research program between 2015 and 2017 using the same methods and protocols.

### 2.1. Participants

Sixty-four adults from three organizations participated in this research. Data from these three organizations were selected as they all had usable e-EMA data. The three organizations included one information and communications technology (ICT) department at a large tertiary education provider that had recently refurbished their traditional office into an activity-based working typology, including nonassigned workstations (some height adjustable), meeting rooms, quiet booths, a central communal kitchen and dining area, and separate meeting rooms, all on the one floor (Organization A); one large financial/consultancy organization in a CBD location with large open-plan office typology spread over one floor with a communal seating/eating area (Organization B); and one university non-health-related faculty that had recently relocated to a new state-of-the-art building following active design principles, with occupants in shared or private offices located in pods of 12 offices spread through a large five-story building with interconnected floors and wide corridors and open spaces (Organization C). All three organizations are located in Sydney, Australia. The participants were recruited to participate in the study as part of studies relating to their office space. Prior to providing consent, they were presented with detailed information about the study and given the opportunity to ask questions about the study.

Ethics approval to collect the data was obtained through the University of Sydney Human Research Ethics Committee under the applications 2013/637 and 2015/107.

*2.2. Protocol*

The commercially available LifeData platform (www.lifedatacorp.com) was used to securely collect data. Participants were instructed to download the free LifeData RealLife Exp application to their smartphones. Participants then downloaded the relevant study survey package, LifePak, to commence participation. The application alerted participants at four random times per day between 9am and 5pm from Monday to Friday to complete a short questionnaire (~1 min). A notification appeared on the participants' smartphones and they tapped on the notification to open the LifeData app and respond to the survey. If a participant did not respond within 5 min of the notification alert, the notification disappeared, and the participant was prompted at another random time later. In total, participants were asked to complete 12 EMA surveys over the 5-day period.

The short survey consisted of 13 questions about posture, musculoskeletal issues, work activity, social interactions, mood, and perceptions of engagement and work performance (Table A1).

*2.3. Analyses*

The data were collected through the secure LifeData online platform, which, in addition to the participants' responses, also recorded information on the time the survey was administered (time-stamped), when the participant responded, and the response time for each question.

We collected the data in a joint database and used exploratory analyses to detect patterns within and between organizations and individuals cross-sectionally as well as over time. We analyzed the data using descriptive statistics and visual analytics with regard to the substantial questions asked in the survey but also with regard to respondents' compliance and their frequency of participation in the survey across our three sites. Repeated observations from the same participant allowed us to capture longitudinal patterns as well as variation within individuals. We used the concurrent responses of distinct behavioral dimensions and perceptions in our e-EMA survey to interrogate select associations between them.

**3. Results**

We enrolled 64 respondents from three separate sites for the study (Table 1). We collected 508 complete survey responses, with eight responses per participant on average. The average response rate to the prompts was 53%, and the average time to fill out the survey was 50 seconds.

**Table 1.** Overview of the participants in the three organizations.

| Organization | Sector | Number of Participants | Gender |
| --- | --- | --- | --- |
| A | Service | 25 | 9% female |
| B | Private consultancy | 19 | 62% female |
| C | University faculty | 20 | 55% female |

*3.1. Response Rates*

Figure 1 shows the number of responses per participant across the three sites. There were three possible scenarios that might have occurred when a user received prompts to respond to the survey: (1) they did not see the prompt or they chose to ignore it (both shown in grey in Figure 1); (2) they indicated in the first screening question that they were not at work and therefore exited the survey at this point (shown in black in Figure 1); (3) they responded to the survey, either providing a full set of 12 responses to all questions or responding incompletely to only some of them (shown in red in Figure 1).

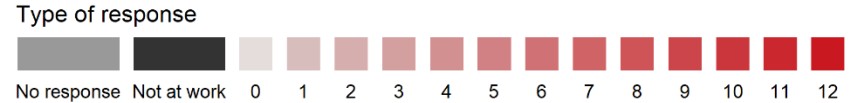

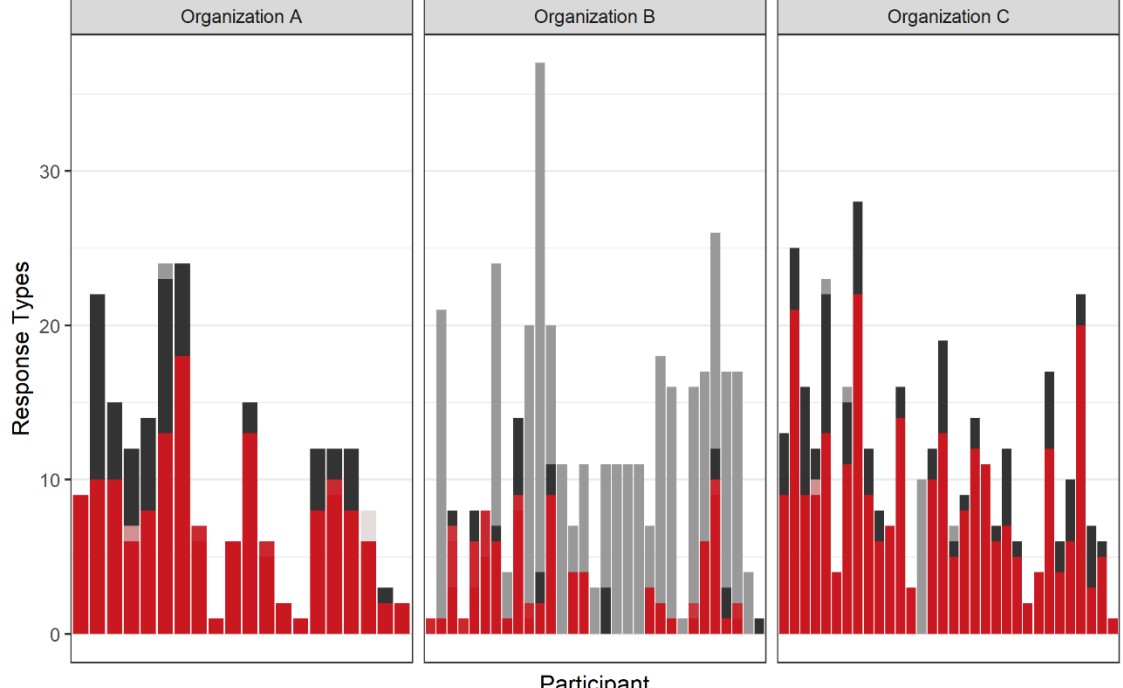

**Figure 1.** Survey app interactions for each participant, with each bar representing a single participant. The height of the bars indicate how often respondents received a query through the app.

Organization A and C showed a similar pattern in that participants largely responded to all the prompts they received, and they completed the survey every time. However, they sometimes indicated they were not at work and therefore exited the survey after the screening question at that instance. In Organization B, the number of nonresponses, i.e., survey prompts that were ignored or dismissed by the respondents, were substantially higher, amounting to more than half. Still, when respondents in this organization chose to interact with the survey, they completed all questions, which was analogous to the other sites.

*3.2. Posture*

Figure 2 illustrates the temporal pattern of type of posture as a proportion of responses in two-hour time slots. Sitting was the most frequently reported posture at all three workplaces (60%–95% across time slots; on average 84%), while standing was reported on average 9% (0%–38%) and walking was reported on average 7% (0%–15%) of the time. At Organization A, standing was high in the morning and again late in the afternoon, and this was inversely related to the proportion of respondents sitting. The other organizations did not show this kind of behavior.

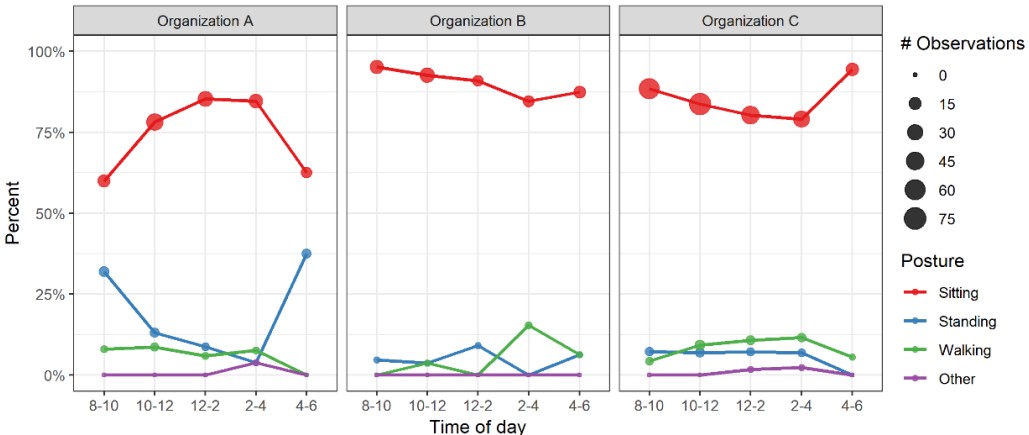

**Figure 2.** Temporal patterns of posture over time. Percentage of respondents from each organization reporting one of four postures (sitting, standing, walking, or other) during two-hour windows throughout the working day. The majority reported sitting throughout the day. The size of points is proportional to the number of observations.

### 3.3. Activities

Further questions about respondents' use of the workplace asked them to specify the activity (work task) they were engaging in at the time. Overall, respondents indicated they were working at their desk 67% of the time. The other most common activities reported were in a meeting (12%), in transit (8%), and working away from their desk (7%). Activities such as taking breaks and other activities, including eating and physical activity, were all reported in less than 3% of survey instances.

The pattern of proportions of these activities were varied throughout the working day as well as between workplaces, although "working at my desk" was consistently the most frequent response across organizations and throughout the day. Figure 3 visualizes the patterns of activity in each organization. The frequency of "working at my desk" varied substantially throughout the day, ranging from 49% to 83% in Organization C, but there seemed to be no consistent patterns between sites.

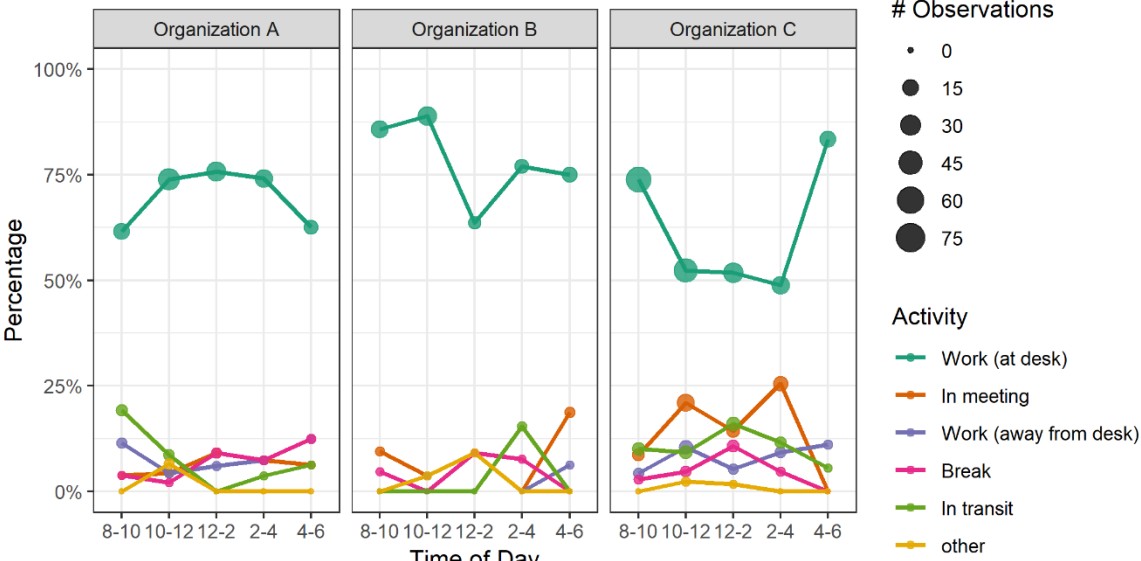

**Figure 3.** Temporal patterns of workplace activity over the course of the day. Percentage of respondents from each organization reporting one of six activities (desk work, meeting, work away from desk, break, transit, and other) during two-hour windows throughout the working day. A majority reported working at their desks throughout the day. The size of points is proportional to the number of observations.

### 3.4. Activities and Social Interactions

Respondents reported to working alone 69% of the time, followed by "with 2–5 others" (11%), "1 other" (10%), and "6 or more other people" (9%). However, e-EMA allowed us to further interrogate the pattern of social interactions and their co-occurrence with work activities. Respondents indicated they were working at their desk alone 55% of all instances. More than half of all meetings (52%) involved six or more participants, while transit was predominantly a solitary activity (77% alone). A break is often an activity that is shared with others, but 44% of responses collected during breaks indicated that the respondent was alone. Remarkably, 15% of work activity at the participants' own desk included other people. Details are illustrated in Figure 4.

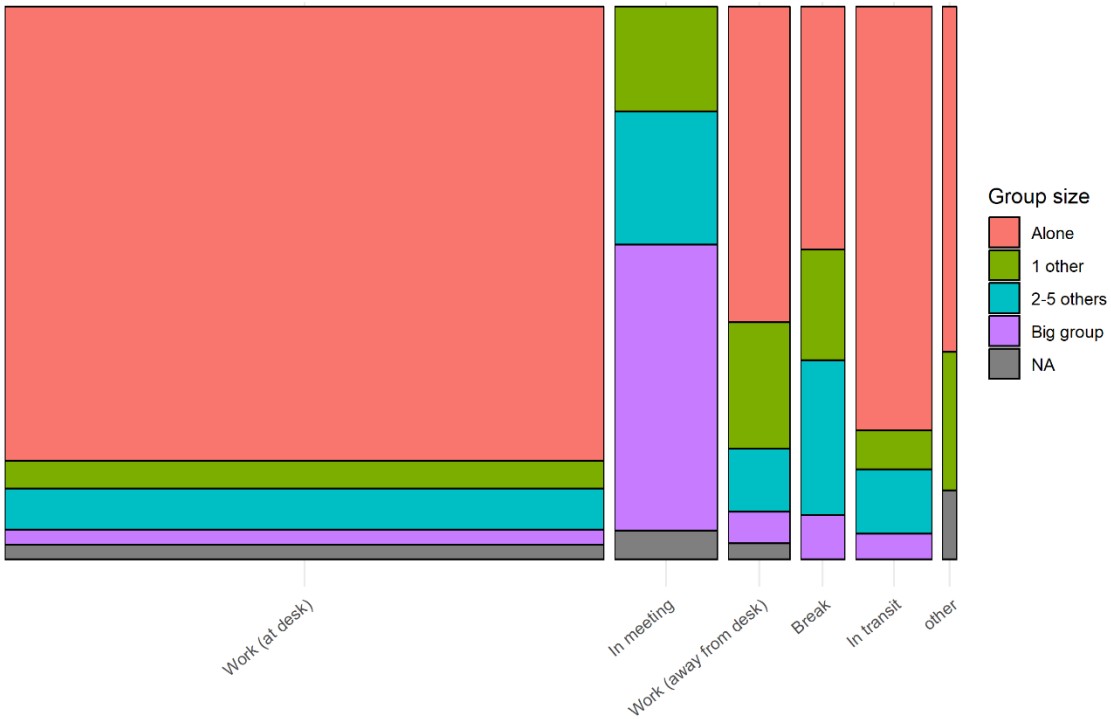

**Figure 4.** Association of work activity and group size. The area represents the proportion of responses for each combination of activity and group size.

### 3.5. Pain and Activities

Figure 5 visualizes the association between activities and perceived musculoskeletal (MSK) pain. Participants who reported experiencing some MSK pain at least once were selected, and their presence or absence of MSK pain were plotted against the activity they were engaging in at the moment of the survey. The findings are presented in Figure 5.

Notably, the incidence rate of pain was highest when "working at a desk", and no respondents reported pain while they were on a break. Additional analysis could not identify any significant association between the time of day and the frequency of musculoskeletal pain; however, it occurred more frequently in later parts of the day.

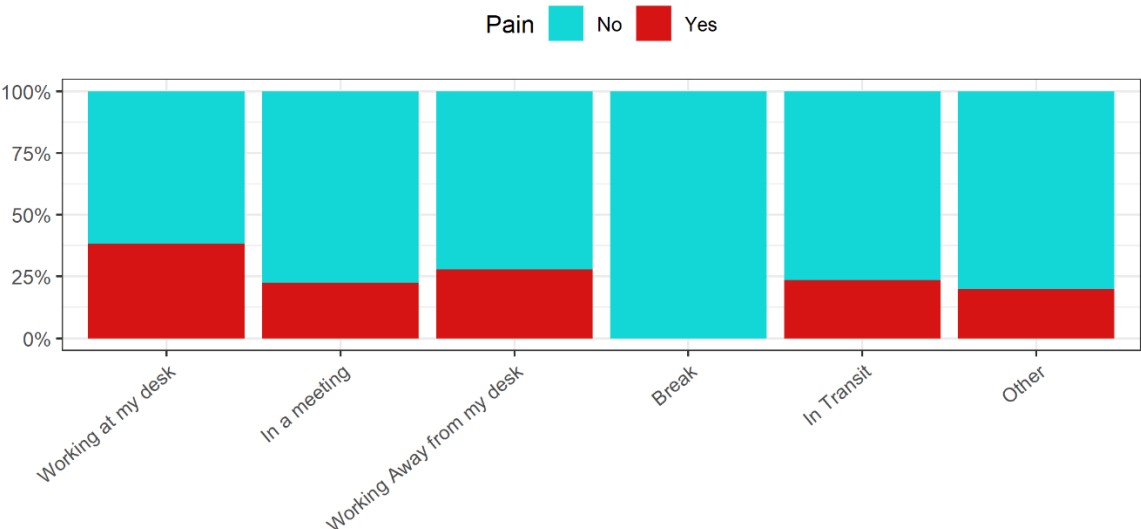

**Figure 5.** Musculoskeletal pain versus activity. Along with their current activity, we asked participants if they experienced any musculoskeletal pain at the moment of the survey. This figure relates the two data points to show how often participants who experienced pain reported it during each activity.

*3.6. Mood and Work Performance*

The e-EMA survey included a self-assessment of mood and work performance across eight different dimensions, including happiness, stress, energy, anxiousness, productivity, motivation, engagement, and creativity. As examples, Figure 6 highlights the respondents' self-assessed productivity and stress. Not only was there a large between-individual variation in the median rating of emotions and work performance, but there was also a large within-individual range in participants' ratings among the responses across measurements over the observation period. The responses within individuals varied between 0 and 9 across time points. There was, however, no discernible pattern for the variation within or between respondents. These findings were consistent across the other mood and work performance measures, as shown in Figure A1.

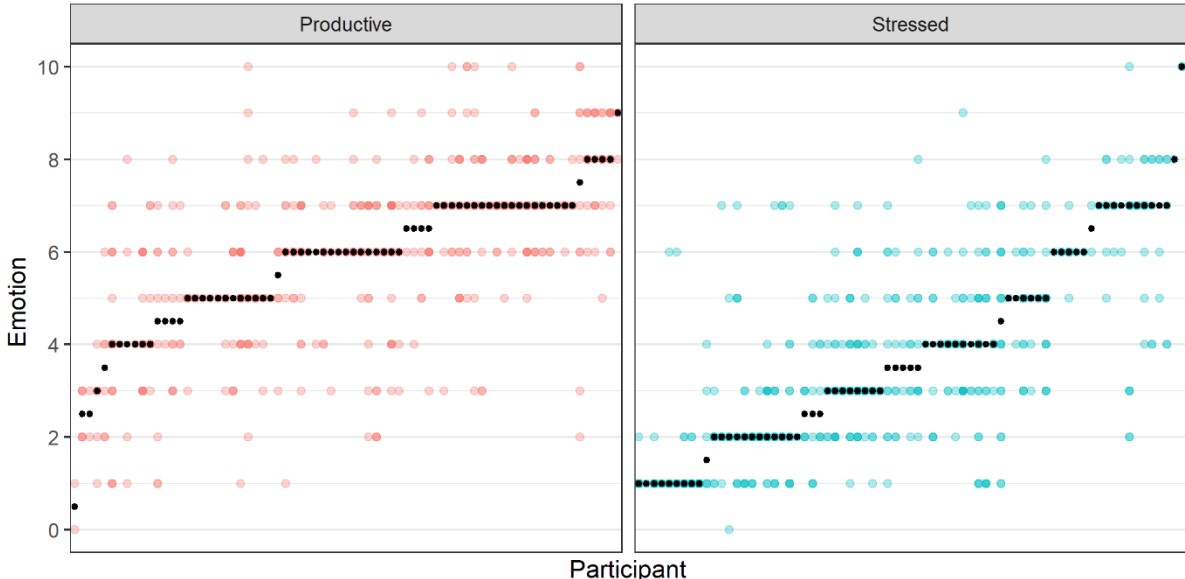

**Figure 6.** Median and range of self-rated perceptions of productivity and stress per participant. Each person is represented vertically, with black points representing their median rating and colored points representing each recorded response.

### 3.7. Mood and Activity

Figure 7 shows the association of current activity with self-reported happiness. For most work activities, the average response was slightly above a neutral value (5), except for responses collected during a break, where happiness was markedly higher. It should be noted that most observations were made while "working at my desk", so the estimates for the other activities may be less accurate due to a smaller sample size.

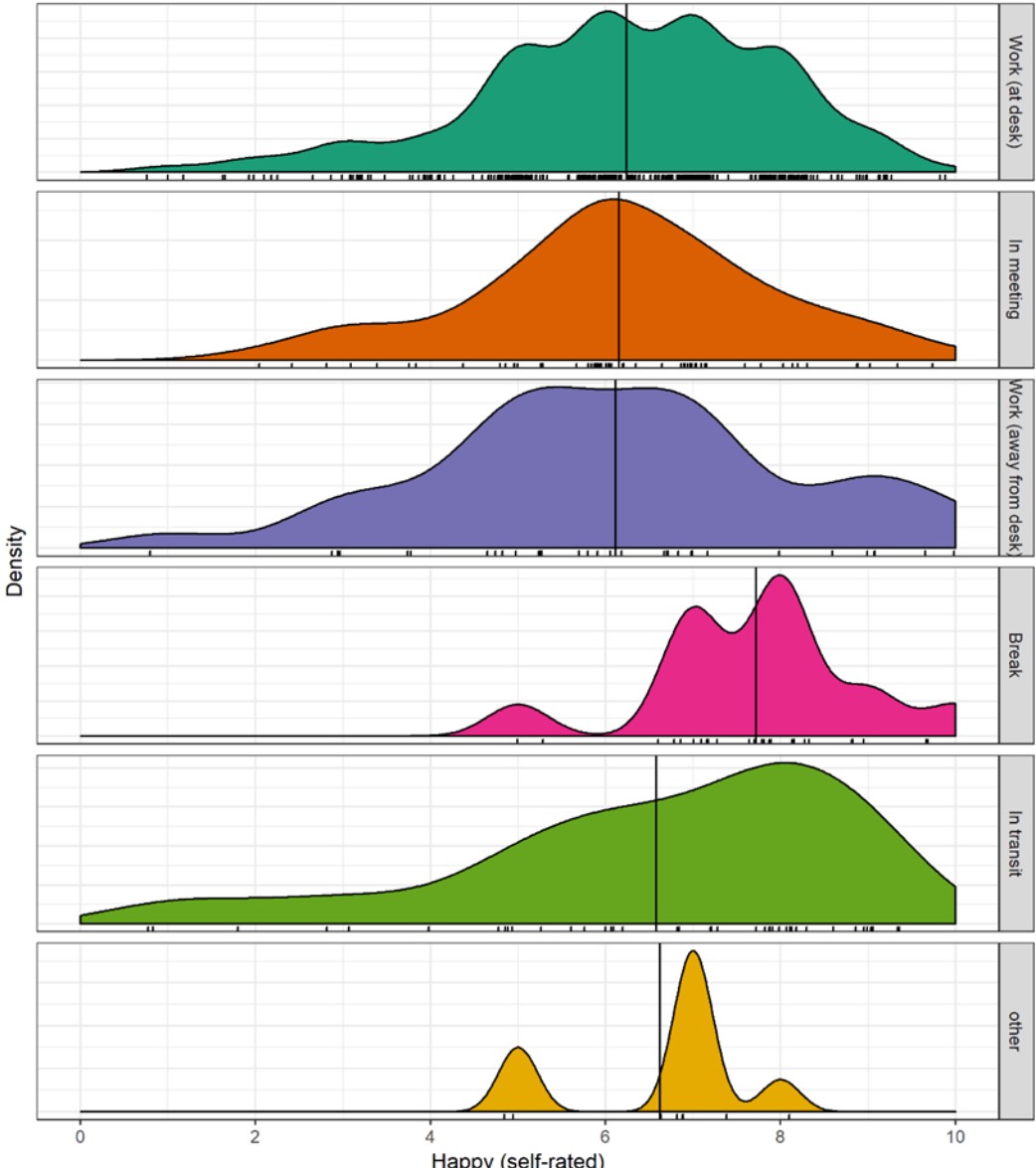

**Figure 7.** Happiness versus activity: distributions of self-rated happiness across all observed activities. The density estimates smooth over the distributions of the possible answers (0–10). The distribution of actual observations is shown as small ticks under each density.

## 4. Discussion

This study was performed with participants in three different office environments. It showed that e-EMA is a useful tool to measure posture, work activity, social interactions, mood, and work performance in the office environment and that it can provide temporal patterns and unique associations due to concurrent information. The benefits of the e-EMA method are the in-the-moment and repeated responses, which have lower recall bias, produce longitudinal information, and allow

measurements of intraindividual variations; low intrusiveness; the concurrent self-reported ratings of a range of variables, which allow a unique understanding of interactions; and the use of existing technology (phones) that people carry with them. The response rate was acceptable (23%–73%; average 53%), especially given the multiple assessments per person, and still provided a large number of data points. Other studies using EMA for data collection have seen response rates between 40% and 89% [12]. e-EMA is a good alternative to standard staff surveys as they are much shorter and can be done from anywhere when the respondent is carrying their phone. This is especially important as many workplaces report that their staff are suffering from "survey fatigue".

The proportion of instances spent sitting, standing, or walking are similar to what other studies have found in office environments using various kinds of measurements, ranging from self-reported estimates [13] to objective measures using inclinometers [14]. This suggests that the EMA method is a valid method to use for estimates of sitting, standing, and walking; however, a well-defined validation study is needed to establish the exact level of validity.

In this study, we found a large within-person variation in mood and work performance measures. We did not find any discernible pattern for the variation within or between respondents' ratings of their mood. This finding suggests that one-off assessments of emotional states are unlikely to reflect well the daily variation within a single respondent. Previous research has reported that many organizational phenomena exhibit substantial within-person variability and are momentary in nature, and it has also been suggested that a considerable proportion of the variability in job performance is attributable to within-person rather than between-person sources [15]. It is therefore important to be able to measure this within-person variation. Our results suggest that the e-EMA measure is sufficiently sensitive and hence suitable in assessing and tracking work performance and mood throughout the workday. This is in accordance with previously reported benefits of EMA as being able to capture meaningful within-person variability, a better understanding of psychological processes, and reduced memory bias [5,10].

In some situations, it may be of value to understand work activities, perceptions, and interactions within a building over an extended time, such as weeks or months, to follow changes carried out or occupants settling in to their new environments. One-off post-occupancy surveys may not be sensitive enough and would hence not be suitable for picking up on these changes. In addition, numerous repeated observations could end up being costly, and it may be hard to receive sufficient follow-up responses due to attrition. e-EMA seems like a good alternative in these situations.

In the vast majority of instances, the participants reported that they were working at their own desk, sitting down, and mostly working alone. Although not surprising, this finding poses questions about the purpose of the workplace and whether work could be designed differently. Could the workplace be used, designed, or fit out in a different manner to better support the tasks that we are engaged in? Could working from home or in co-working hubs closer to home be implemented more effectively for individual tasks? Should staff be encouraged to work more collaboratively while in the office? It has been suggested that the purpose of future offices is to create an experience that will attract staff to spend more time in the workplace, interact, and collaborate [16]. Results from an early 20[th] century study demonstrated that, although not working together, just being co-located in the same workspace increased the mental processes of association and thought [17], highlighting some benefits of being with others while working alone. A recent study by Haynes et al. [18] also showed that social interactions and work interactions had positive associations with perceived impact on productivity. On the flip-side of this, the perceived ability to concentrate well depends on other environmental factors, with high noise levels and other distractions shown to have negative effects [19]. Some reports suggest that work performance tends to be higher for those who telecommute, possibly due to less distractions, more control over time and place of work, and spending some of the time gained by reduced commuting on work [20,21].

This study found no significant difference in mood depending on work activity, social interactions, or time of day, with the exception of people feeling happier in general during break time. This is an

important finding as happy employees have been found to be 12% more productive [22], but being on a break generally means that no traditional work is being performed. The issue is hence how this finding can be used. Is it something about the break from work per se, such as freedom of choice of what to do, that makes people happier? Or is there a way to make more parts of the workday feel like a break? Kim et al. [23] found that microbreaks had significant indirect effects on job performance via positive effect for workers who had lower general work engagement. In addition, in this study, we found no reports of musculoskeletal pain during break time; hence, there is a case for changing settings and taking microbreaks throughout the workday.

Strengths and Limitations

This study has a range of strengths, including multiple assessments per person and participants from a range of workplaces. However, considering they were all from the Australian knowledge or service sector, the results cannot be generalized to other countries and other sectors, such as construction or manufacturing.

This study had a relatively small sample size; however, this is quite common for EMA studies, with researchers often pointing out that the small sample sizes are balanced by the larger number of data points per person [24]. The e-EMA provided a multidimensional picture of the participants' activity and experience in the three workplaces over time.

The moderate response rates (compliance) could possibly be related to the timing of the prompts, with some prompts occurring at inopportune times for the participants, such as during important meetings, transport, or when they had left their phones behind. It is possible that failure to respond to a signal may be correlated with one or more of the variables of interest. Therefore, for example, people may be less inclined to respond under stressful conditions [24], or they may be more inclined to respond when in a positive mood state [10], which introduces possible bias in this study.

In addition, the response rates varied among the participating workplaces. This is likely related to things other than the method itself. At Organization B, we found that the uptake/response rates were low for all research methods used there in previous studies, such as online surveys, objective measures, and focus groups.

Recommendations and Applications

We recommend that a method like e-EMA, which collects information about several aspects of the workplace concurrently and in real time, should be used to get an invaluable understanding of temporal patterns of the workplace; how the space is used; and how activities, social interaction, and work performance are related in the workplace. This information can be used to make improvements to the physical and social workspaces and enhance the occupants' perceptions, mood, and work performance.

## 5. Conclusions

Using e-EMA as a post-occupancy tool in the office setting can provide a detailed understanding of participants' momentary experience. With reduced recall bias and little interruption of the participants' workflow, it allows us to measure concurrently a range of dimensions over time, including activity, social interactions, work performance, and mood. This allows us to better understand within-individual variation throughout the day and over the duration of the study. These interactions are hard to obtain with traditionally used methods. Hence, e-EMA can be used to paint a rich, longitudinal picture of occupants' experiences and behaviors in the offices and buildings they occupy, which in turn provides opportunities to make improvements to the physical and social workspaces and enhance the occupants' work performance and mood based on solid metrics.

**Author Contributions:** The authors contributed to the paper in the following way: conceptualization, L.E.; methodology, L.E. and F.H.; formal analysis, F.H.; Writing—Original Draft preparation, L.E. and F.H.; visualization, F.H.; project administration, L.E.

**Funding:** This research received no external funding.

**Conflicts of Interest:** The authors declare no conflict of interest.

# Appendix A

**Table A1.** Questions and response options of the ecological momentary assessment (EMA) survey.

| Survey Question | Response Options |
|---|---|
| **Are you at work?** | Yes |
| | No |
| **What posture are you in?** | Sitting |
| | Standing |
| | Walking |
| | Lying |
| | Other |
| **Are you experiencing any musculoskeletal pain right now?** | No Pain |
| | I am experiencing pain |
| **What are you currently doing?** | In a meeting |
| | Working at my desk |
| | Working away from my desk |
| | Eating |
| | In transit |
| | On a break |
| | Other |
| | Physical activity |
| | Working remotely |
| **How many people are you doing this with?** | Alone |
| | With 1 other person |
| | With 2–5 other persons |
| | With 6+ other persons |
| **On a scale of 1 to 10, 1 being not at all and 10 being extremely, please rate the following emotions at this moment:** | • Happy<br>• Stressed<br>• Energized<br>• Anxious<br>• Productive<br>• Motivated<br>• Engaged<br>• Creative |

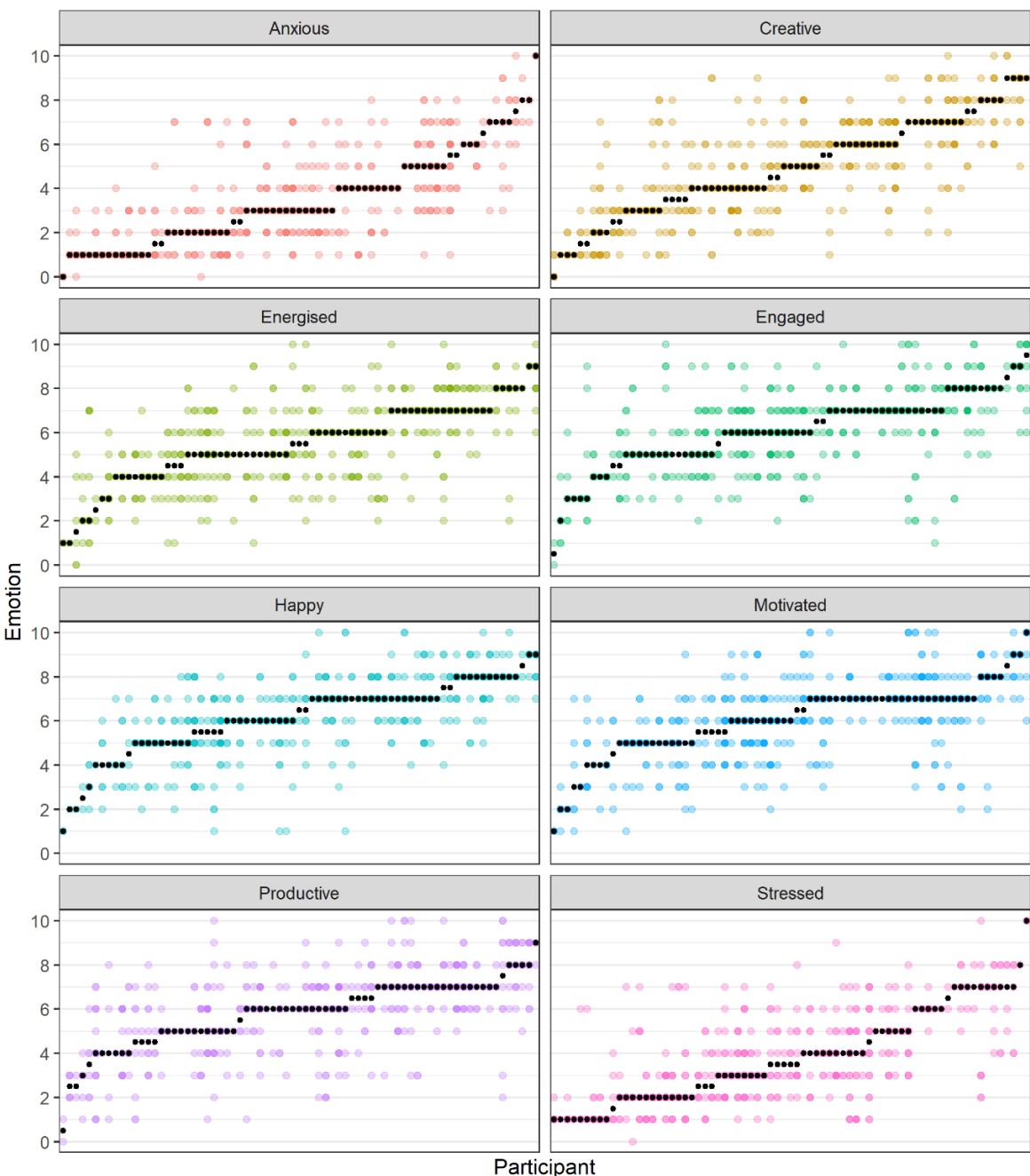

**Figure A1.** Median and range of self-rated perceptions of all emotional states of participants. Each person is represented vertically, with black points representing their median and colored points representing each recorded response.

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
