# Peer review of "Understanding the Office: Using Ecological Momentary Assessment to Measure Activities, Posture, Social Interactions, Mood, and Work Performance at the Workplace"

_buildings, doi:10.3390/buildings9020054_

Round 1

Reviewer 1 Report

An effort is required to the authors, if it is possible to connect the study to the physical places of work. What kind of physical space - work place - are the various case studies associated with the various activities found? (What type of buildings? What kind of internal space distribution, for example vertical connections, collective living spaces, etc.)

Important suggestions could come from the following studies:

Oseland, N., Marmot, A., Swaffer, F., Ceneda, S. (2011). Environments for successful interaction. Facilities, 29 (1), pp. 50-62. Doi: 10.1108/02632771111101322

Smith, L.; Sawyer, A.; Gardner, B. et al. (2018) Occupational physical activity habits of UK office workers: Cross-sectional data from the active buildings study. International Journal of Environmental Research and Public Health, 15 (6), Article number 1214, doi: 10.3390/ijerph15061214

- Why are the results first presented (chapter 2) of the method used (chapter 4)? What is the criterion?

- between lines 25-25 could be cited some examples, there are many of these: headquarters google, microsoft ... etc.

- between lines 45-46 it could be interesting a link with the study method exposed in: Di Fabio A and Bucci O (2016) Green Positive Guidance and Green Positive Life Counseling for Decent Work and Decent Lives: Some Empirical Results. Front. Psychol. 7:261. doi: 10.3389/fpsyg.2016.00261

- lines 199-200; they talk about "perceptions and interactions within a building", very interesting, but should be better specified the type of building, in which kind of spaces.

- lines 205-217 is for me one of the most interesting parts that deserves to be expanded with some more reference.

Author Response

We wish to thank the reviewer for their time and for providing constructive comments.

Please find responses to reviewer's comments below:

                Reviewer 1         

R1.1       An effort is required to the authors, if it is possible to connect the study to the physical places of work. What kind of physical space - work place - are the various case studies associated with the various activities found? (What type of buildings? What kind of internal space distribution, for example vertical connections, collective living spaces, etc.)      

We have now included better descriptions of the physical spaces in the methods section.

R1.2       - Why are the results first presented (chapter 2) of the method used (chapter 4)? What is the criterion?  

This order was specified in the journal's guidelines, however the journal was fine with us changing the paper to an IMRAD organisation. The sequence of sections has been adjusted.

R1.3       between lines 25-25 could be cited some examples, there are many of these: headquarters google, microsoft ... etc.  

That is a good idea: We have now included recent examples of corporate buildings that emphasise the importance of workplace design for employee satisfaction, wellbeing and retention.

R1.4       - between lines 45-46 it could be interesting a link with the study method exposed in: Di Fabio A and Bucci O (2016) Green Positive Guidance and Green Positive Life Counseling for Decent Work and Decent Lives: Some Empirical Results. Front. Psychol. 7:261. doi: 10.3389/fpsyg.2016.00261      

Although the suggested study to reference is very interesting, we do not see clear links with the present manuscript.

R1.5       - lines 199-200; they talk about "perceptions and interactions within a building", very interesting, but should be better specified the type of building, in which kind of spaces.

See response to R1.1

R1.6       - lines 205-217 is for me one of the most interesting parts that deserves to be expanded with some more reference.         

We have now further expanded this section.

Reviewer 2 Report

I have reviewed the article ‘Understanding the office – using Ecological  Momentary Assessment to measure activities,  posture, social interactions, mood, and work  performance at the workplace’. Please address the following aspect to improve the paper.

For a better understanding of the paper, please order the sections be followed: 1 Introduction 2. Method 3 Results 4. Discussion and 5 Conclusions

The survey can be put in the annexes

The literature review conducted is limited, please consider expanding it. MDPI has a lot of papers about health and work.

Please, expand the methods, for example: how and why was selected the organizations?

Author Response

                Reviewer 2         

We wish to thank the reviewer for their time and for providing constructive comments.

Please find responses to reviewer's comments below:

R2.1       For a better understanding of the paper, please order the sections be followed: 1 Introduction 2. Method 3 Results 4. Discussion and 5 Conclusions 

The sequence of sections has now been adjusted to follow the standard IMRAD organization.

R2.2       The survey can be put in the annexes    

The survey has been moved to an Appendix

R2.3       The literature review conducted is limited, please consider expanding it. MDPI has a lot of papers about health and work.           

We expanded the literature review including additional references, especially from MDPI. Added comments on p 1 and 2 on the relation of health on building design, and the use of traditional survey methods in post-occupancy evaluations.

R2.4       Please, expand the methods, for example: how and why was selected the organizations?               

As this paper reports on secondary analysis of data collected in other studies, we have not provided much detail on selection and recruitment. We have however added a few sentences in Methods for some clarification.